# Poor Treatment Outcomes of Locally Advanced Cervical Adenocarcinoma of Human Papilloma Virus Independent Type, Represented by Gastric Type Adenocarcinoma: A Multi-Center Retrospective Study (Sankai Gynecology Study Group)

**DOI:** 10.3390/cancers15061730

**Published:** 2023-03-12

**Authors:** Toshiyuki Seki, Atsumi Kojima, Shinichi Okame, Satoshi Yamaguchi, Aikou Okamoto, Hideki Tokunaga, Shin Nishio, Yuji Takei, Yoshihito Yokoyama, Manabu Yoshida, Norihiro Teramoto, Yoshiki Mikami, Muneaki Shimada, Junzo Kigawa, Kazuhiro Takehara

**Affiliations:** 1Department of Obstetrics and Gynecology, The Jikei University School of Medicine, Tokyo 105-8461, Japan; t_seki@jikei.ac.jp (T.S.);; 2Department of Obstetrics and Gynecology, Iwate Medical University, Yahaba 028-3695, Japan; 3Department of Gynecologic Oncology, National Hospital Organization Shikoku Cancer Center, Matsuyama 791-0280, Japan; 4Department of Gynecologic Oncology, Hyogo Cancer Center, Akashi 673-8558, Japan; 5Department of Obstetrics and Gynecology, Tohoku University, Sendai 980-8574, Japan; 6Department of Obstetrics and Gynecology, Kurume University School of Medicine, Kurume 830-0011, Japan; 7Department of Obstetrics and Gynecology, Jichi Medical University, Shimotsuke 329-0498, Japan; 8Department of Obstetrics and Gynecology, Hirosaki University, Hirosaki 036-8562, Japan; 9Department of Pathology, Matsue City Hospital, Matsue 690-8509, Japan; 10Department of Pathology, National Hospital Organization Shikoku Cancer Center, Matsuyama 791-0280, Japan; 11Department of Diagnostic Pathology, Kumamoto University, Kumamoto 860-8556, Japan; 12Matsue City Hospital, Matsue 690-8509, Japan

**Keywords:** cervical adenocarcinoma, locally advanced, HPV independent, radical hysterectomy, radiotherapy, gastric type

## Abstract

**Simple Summary:**

Cervical adenocarcinomas have been divided into human papilloma virus-associated (HPVa) and -independent (HPVi) tumors in the updated pathological classification issued by the World Health Organization in 2020. However, few studies have investigated the impact of this new classification on the treatment outcomes and prognosis of locally advanced cervical adenocarcinoma. We investigated the treatment outcomes of 103 and 48 patients with locally advanced (stages IB3–IIIC1) HPVa and HPVi cervical adenocarcinomas, respectively. Most patients underwent radical hysterectomy with or without adjuvant therapy for both tumor types. The overall survival time and survival to progression or death were significantly shorter in the HPVi group than in the HPVa group. In particular, patients with parametrial invasion had highly significant differences in survival. Only five patients with HPVi carcinomas received definitive radiotherapy, and all demonstrated a treatment response. HPVi tumors showed poor outcomes with the current treatment strategy.

**Abstract:**

The revised World Health Organization classification of cervical cancer divides adenocarcinomas into human papillomavirus-associated (HPVa) and -independent (HPVi) types; the HPVi type is represented by the gastric type. The treatment outcomes of locally advanced adenocarcinoma (LaAC), based on this classification, are understudied. We investigated the outcomes of patients with HPVa and HPVi LaACs. Data for all consecutive patients with stage IB3 to IIIC1 adenocarcinoma who received treatment at 12 institutions throughout Japan between 2004 and 2009 were retrieved to analyze progression-free and overall survival. Central pathological review classified 103 and 48 patients as having HPVa and HPVi tumors, respectively. Usual- (84%) and gastric- (90%) type adenocarcinomas were the most frequent subtypes. Surgery was the primary treatment strategy for most patients. Progression-free and overall survival of patients with HPVi were worse than those of patients with HPVa (*p* = 0.009 and 0.032, respectively). Subgroup analysis by stage showed that progression-free survival was significantly different for stage IIB. The current surgical treatment strategy for LaACs is less effective for HPVi tumors than for HPVa tumors, especially those in stage IIB.

## 1. Introduction

Although human papillomavirus (HPV) infection is the main cause of cervical cancer, approximately 5–10% of tumors are HPV-independent (HPVi) [1,2]. In 2020, the World Health Organization (WHO) published a new classification of cervical cancer, which uses morphological features to divide cervical adenocarcinoma into HPV-associated (HPVa) and HPVi types [3]. In this classification, the HPVa includes usual type endocervical adenocarcinoma (UEA), mucinous carcinoma NOS, mucinous carcinoma of intestinal and signet-ring cell types, and stratified mucin-producing carcinoma as morphologic variations. The HPVi includes gastric type (GAS), clear cell type, mesonephric type, and endometrioid adenocarcinoma [3].

Although some studies have reported that HPVi adenocarcinoma, in which GAS is the most frequent pathological subtype, has a poor prognosis [4,5,6], only one study relied on pathological classification instead of HPV molecular testing to examine the treatment outcomes for patients with locally advanced cervical adenocarcinomas by implementing the novel WHO classification [7]; this classification divides patients into HPVi and HPVa and compares survival outcomes between the two.

In particular, previous studies have focused on surgically treated patients [6,7]. However, in the clinical setting, locally advanced cervical cancer is treated using either surgery or radiation therapy with or without concurrent chemotherapy, depending on the pathological diagnosis of the biopsy specimen, tumor status, and patient’s background.

Therefore, in this study, we investigated the treatment outcomes of stages IB3 to IIIC1 cervical adenocarcinoma classified as either HPVa and HPVi according to the 2020 WHO guidelines. We used clinical data for all consecutive patients with cervical adenocarcinoma, regardless of treatment method, who were treated at any of the institutions participating in the Sankai Gynecology Study Group (SGSG).

## 2. Materials and Methods

### 2.1. Patients

This multi-institutional retrospective study was named SGSG-015 and approved by the ethics committee or institutional review board of each participating institution. SGSG-015 was registered with the University Hospital Medical Information Network (UMIN 000016285; https://upload.umin.ac.jp/cgi-open-bin/ctr/ctr_view.cgi?recptno=R000018894, accessed on 10 October 2022).

We included patients with stages IB2–IIB (according to the 2008 International Federation of Gynecology and Obstetrics [FIGO] guidelines) non-squamous cell carcinoma cervical cancer who underwent primary treatment between 2004 and 2009 at any of the 12 institutions participating in SGSG-015 throughout Japan. Staging was classified according to the recently updated 2018 FIGO criteria for the analysis. Patients with adenosquamous carcinoma, small cell neuroendocrine carcinoma, or undifferentiated carcinoma, as well as those with tumors extending to the pelvic wall and/or involving the lower third of the vagina and/or hydronephrosis or nonfunctioning kidney (T3), or para-aortic lymph node metastasis, were excluded. Clinical information, including age, stage, initial tumor size, lymphadenopathy and parametrial invasion in imaging studies, treatment method (i.e., type of surgical procedure, radiation protocol, and chemotherapy regimen), pathological information, information regarding recurrence, and prognosis, was collected from the medical records of included patients.

### 2.2. Treatment Methods

Type III radical hysterectomy was the most common surgical treatment performed. Definitive and adjuvant radiotherapy and concurrent chemoradiotherapy were performed at a dose of 45–50 Gy using whole pelvic external beams with X-rays of ≥6 MV in four field box beams in fractions of 1.8–2 Gy given daily five times per week for 5 weeks, with or without intracavitary brachytherapy (high dose-rate brachytherapy). For concurrent chemoradiotherapy, weekly intravenous platinum agents (five–six cycles) were administered concurrently with radiotherapy. The chemotherapeutic regimens used are described in Appendix A.

### 2.3. Central Pathological Review

A central pathological review was performed to confirm diagnoses. Tumor biopsy specimens collected before treatment were reviewed by two board-certified pathologists specializing in gynecological pathology and oncology (NT and YM). The pathologists did not receive any clinical information and were unaware of the original histopathological diagnoses. Diagnosis was made according to the 2020 WHO classification [3], and patients were allocated to the HPVi and HPVa groups based on their diagnoses. Notably, we did not perform molecular testing of HPV in accordance with the WHO classification, which states that HPV molecular testing is not essential and is usually not required for diagnosis because of the strong correlation between HPVa pathogenesis and morphology [3]. Therefore, we performed immunohistochemical staining for p16 and p53 expression to support the diagnosis as described in the WHO classification [3].

### 2.4. Immunohistochemistry

To examine p16 and p53 expression, immunohistochemical staining was performed using the CINtec^®^ Histology Kit (Roche, Basel, Switzerland) for p16 and anti-human p53 protein antibody (clone DO-7) (Dako; Agilent Technologies, Inc., Santa Clara, CA, USA). Formalin-fixed, paraffin-embedded tissue sections (4 μm thick) were incubated for 20 min in a PT Link, Pre-Treatment Module for Tissue Specimens (Dako for deparaffinization, rehydration, and epitope retrieval (preheating and cooling to 65 °C with heat set to 97 °C). Endogenous peroxidase activity was blocked using Dako EnVision FLEX Peroxidase Blocking Reagent before antibody incubation (Dako; Agilent Technologies, Inc.). Immunohistochemistry was performed using the Dako Autostainer Link 48 device with an EnVision^™^ detection and visualization kit (no. K5007), and Dako EnVision DAB + Chromogen Flex substrate buffer (dilution 1 drop/mL) for color development (all from Dako; Agilent Technologies, Inc.).

Immunostained slides were evaluated by two independent pathologists who were blinded to clinical information. Discrepancies were resolved by reevaluating the slides using a multi-head microscope. Immunostaining for p16 was assessed as positive with diffuse staining of epithelial cells (nuclear and cytoplasmic). Immunostaining of p53 was assessed as wild type (patchy basal positivity) or as an aberrant staining pattern (either a strong diffuse expression pattern when >25% of the cells showed strong positive nuclear staining, or a complete absence of staining).

### 2.5. Statistical Analysis

The patients’ background characteristics were compared between the HPVa and HPVi types using Fisher’s exact test and logistic and multiple regression analysis. Survival analysis was performed using the Kaplan–Meier method; univariate and multivariate analyses were performed using the log-rank test and Cox regression hazard model, respectively.

Progression-free survival (PFS) was defined as the duration from initial treatment until disease progression at any location or last contact. Overall survival (OS) was defined as death or last contact.

All *p* values were two-sided, with *p* < 0.05 indicating significance. All statistical analyses were performed using EZR (Saitama Medical Centre, Jichi Medical University, Saitama, Japan; http://www.jichi.ac.jp/saitama-sct/SaitamaHP.files/statmedOSX.html, accessed on 10 October 2022; Kanda, 2012).

## 3. Results

### 3.1. Patient Characteristics

This study included 151 patients from the 12 participating institutions. After a central pathological review of these patients, 103 and 48 patients were classified as having HPVa and HPVi tumors, respectively. The median follow-up period was 2197 days (range, 182–5180 days). Most patients in both groups, 86% with HPVa and 90% with HPVi, underwent surgery for the primary treatment, either with or without neoadjuvant and adjuvant therapies. More patients in the HPVi group were older than 50 years compared with the HPVa group (*p* = 0.003). Stage distribution was also different between the two groups (*p* = 0.01), and more patients in the HPVi group had parametrial invasion compared with the HPVa group (*p* = 0.003) (Table 1). To identify significantly different background factors between HPVa and HPVi tumors, we performed a multiple regression analysis and found that only age was independently associated with the histological type (Appendix A). Radical hysterectomy was predominantly employed in patients who underwent surgery, and almost half the patients received neoadjuvant chemotherapy (Table 1).

The chemotherapeutic regimens employed for concurrent chemoradiotherapy, neoadjuvant chemotherapy, and adjuvant chemotherapy are described in Appendix A. Detailed pathological subtypes are described in Appendix A, showing that UEA was the most common type of carcinoma in the HPVa group (84%), followed by mucinous carcinoma, and not otherwise specified (15%). GAS was the most common subtype in the HPVi group (90%), followed by clear cell carcinoma (four patients, 2%).

In the HPVa group, 69% and 6% of patients with UEA, 80% and 7% of patients with mucinous not otherwise specified (NOS), and 100% and 0% of patients with adenocarcinoma NOS demonstrated positive staining for p16 and aberrant staining for p53, respectively. In the HPVi group only, 16% and 25% of the patients with GAS and clear cell adenocarcinoma demonstrated positive staining for p16, respectively; 40% of patients with GAS showed aberrant staining for p53 (Appendix A).

### 3.2. Survival Outcomes

We compared survival outcomes between the pathological HPVa and HPVi types: the outcomes of patients with HPVi tumors were worse than those of patients with HPVa tumors in terms of both PFS and OS (*p* < 0.001 and 0.010, respectively) (Figure 1a,b). Median PFS and OS were not reached in the HPVa group. In the HPVi group, the median PFS and OS were 29.4 and 87 months, respectively. Univariate analysis of the clinicopathological factors for PFS and OS revealed that age, FIGO stage, lymphadenopathy, parametrial invasion, treatment method, and histological subtype were significantly correlated to PFS and FIGO stage, whereas lymphadenopathy, treatment method, and histological subtype were significantly correlated to OS (Table 2a,b). The significant factors were included in the multivariate analysis. Age, lymphadenopathy, parametrial invasion, treatment method, and histological type were included in the multivariate analysis for PFS; lymphadenopathy, parametrial invasion, treatment method, and histological type were included in the multivariate analysis for OS. Although FIGO stage was identified as a significant factor in the univariate analysis, it was not included in the multivariate analysis because lymph node status and parametrial invasion are both indicators of the FIGO stage. Lymphadenopathy was significantly associated with PFS (hazard ratio [HR], 2.32; 95% confidence interval [CI], 1.38–3.91; *p* = 0.002) and OS (HR, 2.41; 95% CI, 1.34–4.33; *p* = 0.003). Moreover, pathological type HPVi was significantly associated with worse PFS (HR, 2.24; 95% CI, 1.38–3.65; *p* = 0.009) and OS (HR, 1.85; 95% CI, 1.05–3.25; *p* = 0.032). Regarding treatment methods, radiotherapy and concurrent chemoradiotherapy (RT/CCRT) were significantly associated with worse OS (HR, 0.33; 95% CI, 0.17–0.65; *p* = 0.001) (Table 2a,b).

In addition, we analyzed the outcomes according to p16 expression observed by immunohistochemical staining, which is a surrogate marker of HPV infection, and found that p16-negative tumors had worse prognoses for both PFS and OS (*p* = 0.021, 0.043) (Appendix A).

### 3.3. Sub-Group Analysis According to FIGO Stage

Subsequently, we performed subgroup analyses according to the stages of FIGO which were identified, namely stages IB3–IIA, IIB, and IIIC1, for further analysis. The analysis showed that PFS and OS did not differ significantly between the HPVa and HPVi groups in patients with stages IB3–IIA carcinomas (PFS, *p* = 0.061; OS, *p* = 0.47) (Figure 2a,b). However, the PFS was significantly worse in patients in the HPVi group with stage IIB carcinomas than in those in the HPVa group (*p* = 0.019) (Figure 2c). Moreover, OS showed a slight difference between the two subtypes, although the difference was not statistically significant (*p* = 0.0727) (Figure 2d). The survival curves for patients with HPVi carcinomas at stage IIIC1 were slightly lower than those for patients with HPVa carcinomas; however, the difference between the curves was not significant for either PFS or OS (PFS, *p* = 0.095; OS, *p* = 0.276) (Figure 2e,f).

To further investigate the differences in survival curves between patients in the HPVi and HPVa groups with stage IIB carcinomas, we investigated the prognosis of patients who underwent surgery. Patients in the HPVi group had a worse prognosis than those in the HPVa group in terms of both PFS and OS (PFS, *p* = 0.002; OS, *p* = 0.011) (Appendix A).

### 3.4. Outcome of Definitive Radiotherapy against HPVi Tumors

We identified five patients in this cohort with HPVi tumors who received definitive concurrent chemoradiotherapy. As shown in Appendix A, one patient had stage IB3, one had stage IIA, two had stage IIB, and one had stage IIIC1 disease; all of the carcinomas were GASs. Two patients with stage IIB and IIIC1 disease received concurrent chemotherapy. All patients responded to the therapy, with three patients showing a complete response. Recurrence occurred in three patients, two of whom had disease recurrence in the open field of irradiation and one of whom had recurrence within the field. Regarding prognosis, two patients died as a result of disease recurrence (Appendix A).

## 4. Discussion

This study revealed that, under the current management strategy for cervical cancer, locally advanced HPVi adenocarcinomas have a worse prognosis compared with that for HPVa tumors, especially in stage IIB disease.

To establish the etiology associated with the morphological classification of adenocarcinoma, the International Endocervical Adenocarcinoma Criteria and Classification (IECC) was proposed, which classifies adenocarcinoma into HPVa and HPVi based on its morphological features [8]. Although performing molecular HPV testing is reasonable for providing a precise diagnosis of the HPVa and HPVi subtypes, the morphology-based criteria can reportedly differentiate the HPVa from the HPVi subtype with good reproducibility [9,10]. Therefore, in 2020, the WHO updated the classification of cervical cancer by dividing adenocarcinoma into HPVa and HPVi subtypes according to the IECC criteria, which does not require molecular testing [3,9]. Therefore, the pathological diagnoses of HPVa and HPVi in our study were based on morphological features with the support of immunohistochemical staining for p16 and p53 expression.

In several studies examining the relationship between HPV infection and clinicopathological factors and prognosis in cervical cancer, poor prognosis for HPV-negative tumors was reported [11]. Unfortunately, nearly all these studies focused on squamous cell carcinoma, with only a few including patients with adenocarcinoma, and only one specifically focusing on adenocarcinoma [12]. Throughout the last decade—subsequent to GAS, the most common form of HPVi first reported—several groups have confirmed a poor prognosis of the GAS type. In a pivotal study on GAS, Kojima et al. compared the survival outcomes of patients with 2008 FIGO stages IB–IIB cervical adenocarcinomas after surgery. Through histological examinations of GAS and non-GAS tumors, they demonstrated that patients with GAS had worse outcomes than non-GAS patients [4]. Subsequently, Karamurzin et al. reported in their retrospective study that GAS is more likely to be at a higher stage at diagnosis and patients with GAS have a worse prognosis than those with UEA [13]. Nishio et al. retrospectively analyzed 95 and 233 patients with 2008 FIGO stage IA–IIB GASs and UEAs, respectively, who underwent surgery. They reported that GAS is more likely to have bulky mass, deep stromal invasion, lymphovascular space invasion, parametrial invasion, ovarian metastasis, positive ascitic fluid cytology, and pelvic lymph node metastasis. These characteristics lead to poor outcomes for patients with GAS [6]. Stolnicu et al. also reported that HPVi-type adenocarcinoma is negatively associated with OS and recurrence-free survival in patients with stage IB1 to IB3 disease who undergo surgery [14]. In addition, Cho et al. reported on a single institutional retrospective analysis of patients with stage IB–IIIC2 HPVi and HPVa adenocarcinomas who underwent surgery. The study found that patients with HPVi tumors—71.4% of which were GAS—were more likely to have positive resection margins and higher rates of local recurrence and distant metastasis than those with HPVa tumors [7]. In line with these studies, we found that patients with HPVi tumors of the GAS type had poorer outcomes compared with patients with HPVa tumors under the current treatment strategy. Furthermore, we identified a difference in survival between p16-positive and p16-negative tumors, which is reasonable as p16 status is highly correlated to HPV status [9].

However, unlike the previous studies, in order to analyze clinical data, we included consecutive patients who received primary treatment during the study period, irrespective of the treatment method. Treatment guidelines recommend concurrent chemoradiotherapy to treat locally advanced disease [15]; however, previous retrospective studies only examined patients who underwent surgery, which may have led to biased results, for example, by only reflecting results from patient cohorts that could tolerate surgery. Since surgery is predominantly employed for adenocarcinoma disease in Japan, even for patients with stage IB3–IIIC1 tumors, our study also mainly included surgically treated patients. By including all consecutive patients treated during the study period, we were able to provide actual data that shows the poor prognosis of locally advanced HPVi adenocarcinoma, irrespective of the treatment administered [16].

Moreover, most previous reports included early-stage disease and showed poor outcomes for HPVi tumors [5,6]. In contrast to these studies, our study focused on later-stage IB3–IIIC1 tumors. Regarding the reported outcome of locally advanced disease, Nishio et al. found no difference in the PFS and OS of patients with GAS and UEA in the subgroup of pT1b2 or higher. In contrast, we showed that HPVi tumors in stages IB3 to IIIC1, which closely corresponds to the subgroup pT1b2 or higher, have a worse treatment outcome than HPVa tumors do. It is difficult to determine the exact reason for the conflicting results between the two studies. However, in the subgroup analysis for stage IIB and IIIC1 in our study, there was an apparent survival difference between HPVa and HPVi tumors in stage IIB but few differences in stage IIIC1. In contrast, the Nishio et al. study only considered the pT stage and did not discriminate patients with lymph node metastasis in the subgroup analysis for locally advanced tumors. Therefore, we speculate that patients with HPVi tumors have a worse prognosis than those with HPVa stage IIB tumors without lymph node involvement. However, the prognosis of patients with HPVi is as poor as that of patients with stage IIIC1 HPVa, although stage IIIC1 can include miscellaneous tumors, such as tumors with and without parametrial invasion, large and small tumors, and multiple and single lymph node metastasis, making analysis difficult. These results suggest that the treatment strategy used in the reported studies, such as radical surgery, may not be optimal for stage IIB and IIIC1 HPVi tumors and IIIC1 HPVa tumors.

In our study, we also showed that RT/CCRT seemed to be inferior to surgery in terms of OS in patients with locally advanced cervical adenocarcinoma. The radiosensitivity of cervical adenocarcinoma remains controversial [17,18]. Some studies reported that adenocarcinoma had low radiosensitivity, in line with our data [17]. However, providing a conclusive result on this issue was difficult in this study because of the small number of included patients who received RT/CCRT and the differences in patient background characteristics between patients who received RT/CCRT and patients who received surgery.

We reported the treatment outcomes of radiation therapy for HPVi (GAS) tumors. Harima et al. reported that patients with HPV-negative tumors showed worse OS and disease-free survival than those with HPV-positive tumors among patients who received definitive radiotherapy 20 years ago [19]. However, this study included mainly tumors at stages III and IV, only included three HPV-negative adenocarcinoma patients, and did not perform adenocarcinoma-specific analysis. Therefore, we believe that ours is the first report to describe in detail the outcome of definitive concurrent chemoradiotherapy used to treat HPVi (GAS) tumors. Only five patients in our study received radiotherapy, making it difficult to determine the effectiveness of this method. However, it is noteworthy that all five patients responded to radiotherapy.

This study had several strengths. First, we collected data from all consecutive patients with adenocarcinoma treated at the participating institutions during the study period, thereby enabling the analysis of clinical data that, for example, included patients with poor performance status. Second, a central pathological review was performed to confirm the diagnoses by two experienced gynecologic pathologists (NT and YM). Third, we showed the treatment response of locally advanced HPVi with a relatively larger number of patients than that of previous reports. Fourth, we demonstrated the response of GAS HPVi to concurrent chemoradiotherapy, which has not been previously documented.

However, this study had some limitations. First, the number of patients with HPVi carcinomas included in this study was smaller than that of patients with HPVa carcinomas because HPVi tumors are rare. Second, since few cases of HPVi tumors other than GAS were included in this study, the results of this study do not reflect the features of other rare types of HPVi tumors, such as clear cell adenocarcinomas, and endometrioid, serous, and mesonephric tumors. Third, almost half the patients received neoadjuvant chemotherapy, which might have affected our results.

## 5. Conclusions

This study used clinical data and demonstrated that locally advanced HPVi adenocarcinomas have a worse prognosis than HPVa tumors with the current treatment strategy for cervical adenocarcinoma, especially in patients with stage IIB disease. Novel strategies for treating HPVi tumors are required.

## Figures and Tables

**Figure 1 cancers-15-01730-f001:**
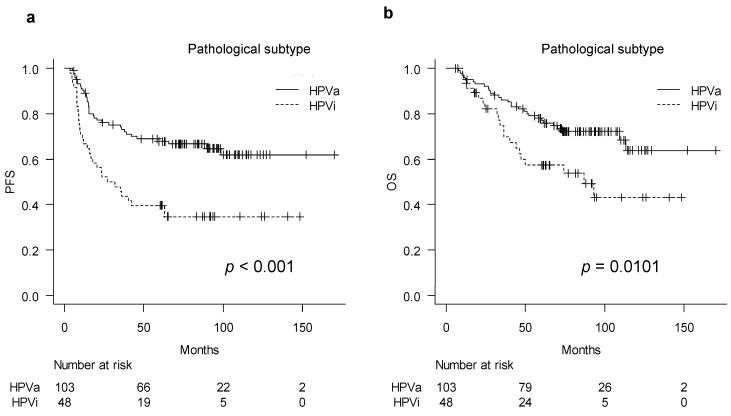
Survival curves of HPV-associated and HPV-independent tumor: (**a**) Kaplan–Meier curve describing progression-free survival and (**b**) overall survival of each group. HPVa, human papilloma virus-associated; HPVi, human papilloma virus-independent; PFS, progression-free survival.

**Figure 2 cancers-15-01730-f002:**
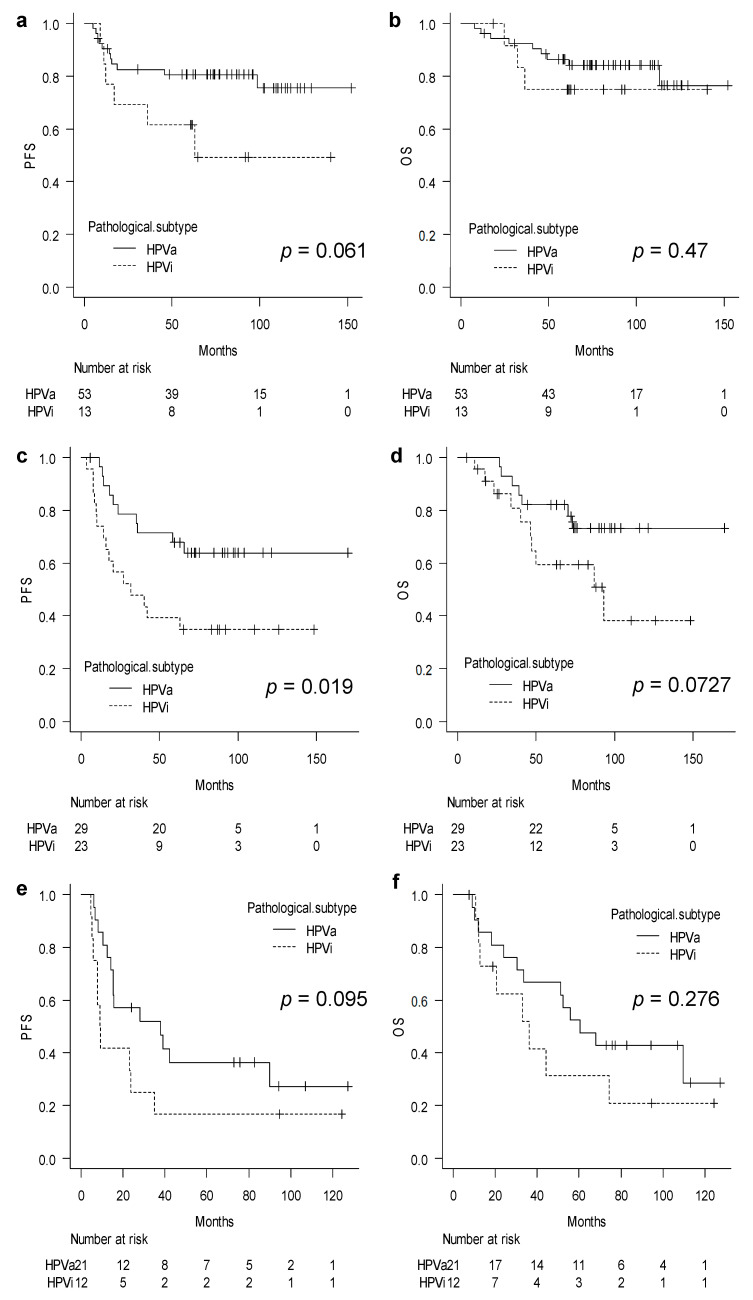
Sub-group analysis stratified by FIGO stage: (**a**) Progression-free survival in stages IB3–IIA; (**b**) overall survival in stages IB3–IIA; (**c**) progression-free survival in stage IIB; (**d**) overall survival in stage IIB; (**e**) progression-free survival in stage IIIC1; and (**f**) overall survival in stage IIIC1. FIGO, International Federation of Gynecology and Obstetrics; HPVa, human papilloma virus-associated; HPVi, human papilloma virus-independent; OS, overall survival; PFS, progression-free survival.

**Table 1 cancers-15-01730-t001:** Patient characteristics.

Characteristic		HPVa Group	HPVi Group	*p* Value
		n = 103	n = 48	
Age	≤50	62 (60.2)	16 (33.3)	0.003
	>50	41 (39.8)	32 (66.7)	
FIGO stage	IB3	41 (39.8)	7 (14.6)	0.01
	IIA	12 (11.7)	6 (12.5)	
	IIB	29 (28.2)	23 (47.9)	
	IIIC1	21 (20.4)	12 (25.0)	
Lymphadenopathy	-	85 (82.5)	35 (72.9)	0.197
	+	18 (17.5)	13 (27.1)	
Parametrium invasion	-	73 (72.3)	22 (45.8)	0.003
	+	28 (27.7)	26 (54.2)	
Tumor diameter	>40 mm	18 (17.4)	16 (33.3)	0.07
	≤40 mm	81 (78.6)	30 (62.5)	
	Unknown	4 (3.9)	2 (4.2)	
Treatment method	RT/CCRT	14 (13.6)	5 (10.4)	0.793
	Surgery	89 (86.4)	43 (89.6)	
Surgical method	RH	79 (76.7)	38 (79.2)	0.036
	MRH	1 (1.0)	3 (6.2)	
	TAH	0 (0.0)	1 (2.1)	
	Unknown	9 (8.7)	1 (2.1)	
NAC	-	57 (55.3)	34 (70.8)	0.077
	+	46 (44.7)	14 (29.2)	
Adjuvant therapy	-	30 (29.1)	7 (14.6)	0.194
	CCRT	22 (21.4)	21 (43.8)	
	RT	14 (13.6)	3 (6.2)	
	CT	36 (35.0)	16 (33.3)	
	RT + CT	1 (1.0)	1 (2.1)	

HPVa, human papilloma virus-associated; HPVi, human papilloma virus-independent; FIGO, International Federation of Gynecology and Obstetrics; NAC, neoadjuvant chemotherapy; RT, radiotherapy; CCRT, concurrent chemoradiotherapy; RH, radical hysterectomy; MRH, modified radical hysterectomy; THA, total abdominal hysterectomy; CT, chemotherapy.

**Table 2 cancers-15-01730-t002:** Analysis of prognostic factors by univariate and multivariate analysis: (a) progression-free survival and (b) overall survival.

(a)					
		Univariate		Multivariate	
		HR (95% CI)	*p* Value	HR (95% CI)	*p* Value
Age	≥50	1	0.044	1	0.5
	<50	1.66 (1.01–2.71)		1.22 (0.69–2.15)	
FIGO Stage	IB3 IIA	1	>0.001		
	IIB IIIC1	2.73 (1.57–4.75)			
Lymphadenopathy	-	1	0.002	1	0.003
	+	2.32 (1.38–3.91)		2.17 (1.29–3.66)	
Parametrium invasion	-	1	0.023	1	0.24
Parametrium invasion	-	1	0.023	1	0.24
Tumor diameter	>40 mm	1	0.51		
	≤40 mm	0.83 (0.48–1.44)			
Treatment method	RT/CCRT	1	0.028	1	0.099
	Surgery	0.49 (0.26–0.93)		0.54 (0.28–1.03)	
NAC	-	1	0.31		
	+	0.77 (0.47–1.28)			
Histological type	HPVa	1	>0.001	1	0.009
	HPVi	2.47 (1.52–4.01)		2.24 (1.38–3.65)	
(**b**)					
		Univariate		Multivariate	
		HR (95% CI)	*p* value	HR (95% CI)	*p* value
Age	≥50	1	0.15		
	<50	1.51 (0.86–2.65)			
FIGO Stage	IB3 IIA	1	0.001>		
	IIB IIIC1	3.03 (1.58–5.81)			
Lymphadenopathy	-	1	0.003	1	0.006
	+	2.41 (1.34–4.33)		2.30 (1.27–4.13)	
Parametrium invasion	-	1	0.15	1	0.65
	+	1.51 (0.86–2.64)		1.14 (0.64–2.04)	
Tumor diameter	<40 mm	1	0.96		
	≤40 mm	0.98 (0.51–1.89)			
Treatment method	RT/CCRT	1	0.001>	1	0.001
	Surgery	0.31 (0.16–0.6)		0.33 (0.17–0.65)	
NAC	-	1	0.22		
	+	0.69 (0.39–1.25)			
Histological type	HPVa	1	0.012	1	0.032
	HPVi	2.07 (1.18–3.63)		1.85 (1.05–3.25)	

PFS, progression-free survival; OS, overall survival; HR, hazard ratio; CI, confidence interval; FIGO, International Federation of Gynecology and Obstetrics; NAC, neoadjuvant chemotherapy; RT, radiotherapy; CCRT, concurrent chemoradiotherapy; HPVa, human papilloma virus-associated; HPVi, human papilloma virus-independent.

## Data Availability

The data presented in this study are available on request from the corresponding author. The data are not publicly available as they were privately collected.

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
