# Peer review of "Poor Treatment Outcomes of Locally Advanced Cervical Adenocarcinoma of Human Papilloma Virus Independent Type, Represented by Gastric Type Adenocarcinoma: A Multi-Center Retrospective Study (Sankai Gynecology Study Group)"

_cancers, 2023, doi:10.3390/cancers15061730_

Round 1

Reviewer 1 Report

Due to the importance of the presence of HPV in response to treatment, the presented work is an important contribution, especially since it concerns a different group of cancers than squamous. Center cooperation made it possible to gather a larger group, and thus it is possible to select important factors that could potentially influence the application of more aggressive treatment. After reading the article, I noticed in particular the lack of a detailed description of the HPV test and the statistical analysis. Given that this is a retrospective, multicenter study with different treatment regimens, statistics are very difficult, so I am asking all authors to discuss the "statistical treatment scheme". Also, please update the results.

Below are detailed suggestions:

(1) The basic criterion for the analysis is the classification of tumors as HPVi or HPVa. As indicated in the review by Fernandes et al. \(Fernandes et al. Human papillomavirus-independent cervical cancer. Int J Gynecol Cancer. 2022 Jan;32(1):1-7. doi: 10.1136/ijgc-2021-003014)., they are based on molecular technology. Meanwhile, in the article there is no mention on the basis of which methodology the classification was made since the differential diagnosis includes the HPV test. The following are desirable in the diagnosis of HPVa/HPVi: p16 overexpression; HPV detection; negative ER, PR, and (usually) vimentin; wildtype p53. How does Table S2 relate to this? The exact method of HPV molecular testing should be presented. It is especially important that the research was carried out on specimens from 2004-2009. It is also important to include people who conducted these studies in the group of authors, because molecular research requires skillful classification (similarly to qualifying for a histological type - which results in the placement of pathologists).

(2) If possible, specifying the number of copies for the HPV group would be advisable. Mainly because small amounts of HPV may only be of importance as a co-factor and not a major factor in carcinogenesis.

(3) Table S2: I'm guessing that p16/p53 was done out of 103, the number N should indicate the true number of p16 or p53 tests done (first line 64, not 87)??? It seems that in the remaining sections p16 was not marked???? It was written: “In the HPVa group, 71% of cases had staining that was positive for p16 and 4% showed aberrant staining for p53. Only 15% of cases in the HPVi group had positive staining for p16 and 40% showed aberrant staining for p53”. Meanwhile: In the HPVa group there are 60+12+1+4 (77 in total), of which 73 (60+12+1) are p16+, which gives 94.8%; and 5.2% (4 cases per 77) are p53 abnormal. If my calculations are wrong, it is due to incorrectly constructed Table S2 (no information/columns with numbers p16- and p53 normal).

(4) Line 71-72 “few studies have examined the treatment outcomes of the recently classified HPVi and HPVa types in patients with locally advanced cervical adenocarcinomas - please specify (citations) who did this "few" research

Line 80 (or 231): what is "real-world data"? some other expression?.

(5)Table 1: Please use a multiple regression model to identify independent significant factors  differentiating between HPVa and HPVi.

Line 156-7: “More patients in the HPVi group were older than 50 years than were patients in the 157

HPVa group (p = 0.003)” for such conclusions it would be advisable to calculate OR, which is 3 in your results. Line 158-9:  “More patients in the HPVi group had parametrial invasion than did those in the HPVa group (p = 0.003)”. More patients in the HPVa group (72.3%) had parametrial invasion “minus” than did those in the HPVi group. OR calculation will help in interpretation. Line 159-60 The FIGO stage was more likely to be higher in the HPVi group than in the HPVa group (p = 0.01). Which of IIA, IIB, IIIC1 was named "higher"? What was compared to what? Combine the lower stages with the higher ones to calculate the OR (e.g. 35/13 to 50/53 = 2.8 OR - which means that the incidence of IIB/IIIc1 to IB3/IIA is 2.8 higher in HPVi than HPVa). For OR representations, logistic regression is also a good solution.

Table S1. Add a row below the table and summarize the cases in columns.

(6) Table 2: Please provide (add) the univariate cox for all parameters from Table 1 including HPV status (or omit those with an unreliable group such as surgical method), and then present the multivariate cox results in the next columns. The reason for doing so is: it must be clear which parameters have been taken into account for the multivariate analysis. Then, multiple analysis present which ones are left/affect the OS or PFS.

(7) Line 206: lost zero before seven in sentence: “OS also showed a possible difference between the two subtypes, although the difference was not statistically significant (p = 0.727) (Figure 2(d)”.

(8) In the discussion, I will ask you to focus on those articles that will help to understand the results.

Line 233: “Additionally, we report here that HPVi GASs respond to definitive radiotherapy” This should be confirmed statistically.

Reviewer 2 Report

The authors investigated the treatment outcomes and prognosis of 151 patients with locally advanced HPV-associated (HPVa) and -independent (HPVi) cervical adenocarcinoma. This was a multi-center (12 institutions) retrospective study throughout Japan. They concluded that the overall survival time and survival to progression or death were significantly shorter in HPVi group than in the HPVa group.

The most important point of this article is pathological diagnosis of the subclassification of cervical adenocarcinoma. They collected relatively rare histopathological HPVi cervical adenocarcinoma of 48 cases, represented by gastric type adenocarcinoma carcinomas, Especially, five patients with HPVi carcinomas received definitive radiotherapy, and all demonstrated a treatment response.

This article is informative for gynecologists and gynecological pathologists and other researchers.

It might be improved on the following issues.

1.    Introduction, P.2, line 64: In this study, were there any mixed type subclassification of cervical adenocarcinoma?

2.    Materials and Methods, P.3, line 114: In central pathological review, how many pathologists specializing in gynecological pathology and oncology?

3.    P.5, line 177 survival outcomes and P.6, Table 2: Was survival analysis performed between immunohistochemically positive and negative adenocarcinoma cases for p16?

Round 2

Reviewer 1 Report

Thank you for your responses,

I see a significant improvement.

However, I have serious doubts about the statistics and suggest asking for help from a statistician. 

OR should be shown in Table 1, because it is used for univariate analysis of dichotomous data. 

In contrast, OR is not used to represent multivariate analysis. In multivariate regression, the beta coefficient is very important (in addition to the P value).

Improvement required.
